# Definitional Challenges in Understanding Hypertrophic Cardiomyopathy

**DOI:** 10.3390/diagnostics14222534

**Published:** 2024-11-13

**Authors:** Jan M. Federspiel, Jochen Pfeifer, Frank Ramsthaler, Jan-Christian Reil, Peter H. Schmidt, Vasco Sequeira

**Affiliations:** 1Institute for Legal Medicine, Faculty of Medicine, Saarland University, Campus Homburg, Building 49.1, Kirrberger Straße 100, 66421 Homburg/Saar, Germany; 2Department for Pediatric Cardiology, Saarland University Medical Centre, Building 9, Kirrberger Straße 100, 66421 Homburg/Saar, Germany; 3Department of General and Interventional Cardiology, Heart and Diabetes Centre North Rhine-Westphalia, Ruhr University Bochum, 32545 Bad Oeynhausen, Germany; 4Department for Translational Research, Congestive Heart Failure Centre, University Clinic Wuerzburg, Building A15, Am Schwarzberg 15, 97078 Wuerzburg, Germany; sequeira_v@ukw.de

**Keywords:** hypertrophic cardiomyopathy, obstructive hypertrophic cardiomyopathy, macroscopic pathoanatomy, disease definition, myocardial structural alterations

## Abstract

Hypertrophic cardiomyopathy (HCM) is the most common hereditary cardiomyopathy. It is often caused by mutations of genes encoding for sarcomeric or sarcomere-associated proteins. Despite its clinical importance, divergent definitions are published by major cardiology societies. Some regard HCM as a specific genetic disease, whereas others define it as a broad ‘spectrum of the thick heart’. The present narrative review aimed to assess both definitions from a pathoanatomical perspective. As a conjoint interdisciplinary and translational approach is needed to further increase knowledge and improve the understanding of HCM, the PubMed database was searched using several advanced search algorithms to explore the perspectives of the (forensic) pathologist, clinician, and basic researcher regarding the difference between the definitions of HCM. This discrepancy between definitions can impact critical data, such as prevalence and mortality rate, and complicate the understanding of the disease. For example, due to the different definitions, research findings regarding molecular changes from studies applying the narrow definition cannot be simply extended to the ‘spectrum’ of HCM.

## 1. Introduction

In 1944, Levy and Von Glahn described the clinical and pathological features of what they termed ‘cardiac hypertrophy of unknown cause’ [1,2]. Four years later, Evans recognized the familial occurrence of the disease and named it ‘familial cardiomegaly’ [2,3]. In 1957, Teare detailed the morphology of the condition as ‘asymmetrical hypertrophy of the heart in young adults’ [2,4]. By 1959, Braunwald and Morrow had identified dynamic obstructions of the outflow tract (OT, summary of abbreviations) of the left ventricle (LV) [2,5]. In 1961, Paré et al. recognized the autosomal-dominant inheritance pattern of the disease [2,6]. Given the marked involvement of the interventricular septum (IVS) [7,8], the disease was also historically referred to as ‘asymmetric septal hypertrophy’ [9] or ‘idiopathic hypertrophic subaortic stenosis’ [10]. In 1990, Geisterfer-Lowrance et al. identified a mutation in the myosin heavy chain as the cause of ‘familial hypertrophic cardiomyopathy’ [11]. Today, HCM is considered the most prevalent genetic cardiac disease [7,12] and a global health burden [13]. Its high prevalence, estimated to range from 1 to 3 in 500 people, is persistently reported in preclinical (basic) studies [14], research compendia [12], and clinical guidelines [8,15,16]. HCM research has shown that the condition is characterized by more than just a ‘thickened heart’. It is also associated with hypercontractility [17] and diastolic dysfunction, which precedes myocardial hypertrophy [7,8]. However, a single unified definition of HCM remains elusive despite these advances.

The European Society of Cardiology (ESC) implemented a phenotypic approach into their 2014 guidelines that mainly focused on the criterion of a ‘thick heart’ to define the disease [16] based on a phenotypic and clinically, respectively practically oriented classification of cardiomyopathies presented in 2008 [18]. This approach has also been used in the most recent ESC guidelines on cardiomyopathies from 2023 [8] and is termed the ‘phenotypic approach to cardiomyopathies’ [8]. Some authors describe this definition as an ‘umbrella diagnosis’ [19]. In contrast, the American Heart Association (AHA) applies a narrower definition that emphasizes HCM as a genetic disease [15]. These differences can be significant, although they seem subtle; for example, non-uniform definitions of (sudden) cardiac death (SCD) have hampered epidemiological studies [20]. (The historical development regarding HCM and its definitions are illustrated in Figure 1). However, the definitional gap between the AHA and ESC is not adequately emphasized in the scientific literature to the best of the authors’ knowledge. The present narrative review is aimed at raising awareness about the nuances and fundamental differences in the definitions of HCM applied by the two major cardiology societies. First, both definitions are outlined. Second, the pathology of a hypertrophic heart is described with a focus on the thickened ventricular wall, which is often the first observable finding in both clinical and (forensic) pathological settings. Consequently, the review explores the definitional gap from this pathological perspective, and it does not advocate for one definition over the other.

## 2. Materials and Methods

To prepare this narrative review, the PubMed database was searched using several advanced search algorithms to explore the different perspectives of basic researchers, clinicians and (forensic) pathologists regarding the difference between the definitions of HCM (literature search detailed in Appendix A).

## 3. HCM Definitions by ESC and AHA

The ESC defines HCM as “the presence of increased LV wall thickness (with or without RV [(right ventricular)] hypertrophy) or mass that is not solely explained by abnormal loading conditions” ([8] page 12, citing [18]). This broad definition, which can be described as an ‘umbrella diagnosis’ [19], captures a spectrum of conditions characterized by the symptom of the ‘thick heart’. This includes thick heart phenocopies such as glycogen storage diseases (e.g., glycogen storage disease type IIIa) [21], mitochondrial cardiomyopathies [21] (e.g., Barth syndrome [22]), lysosomal storage disorders (e.g., Fabry disease), and multisystemic disorders such as Noonan syndrome [21] (for details, see references [8,21,22]).

In contrast, the AHA defines HCM as “a disease state in which morphologic expression is confined solely to the heart. It is characterized predominantly by LV [hypertrophy] in the absence of another cardiac, systemic, or metabolic disease capable of producing the magnitude of hypertrophy evident in a given patient and for which a disease-causing sarcomere (or sarcomere-related) variant is identified, or genetic etiology remains unresolved” ([7] page e1247). This narrower definition focuses specifically on HCM as a genetic myocardial disorder rather than a spectrum of diseases.

**Figure 1 diagnostics-14-02534-f001:**
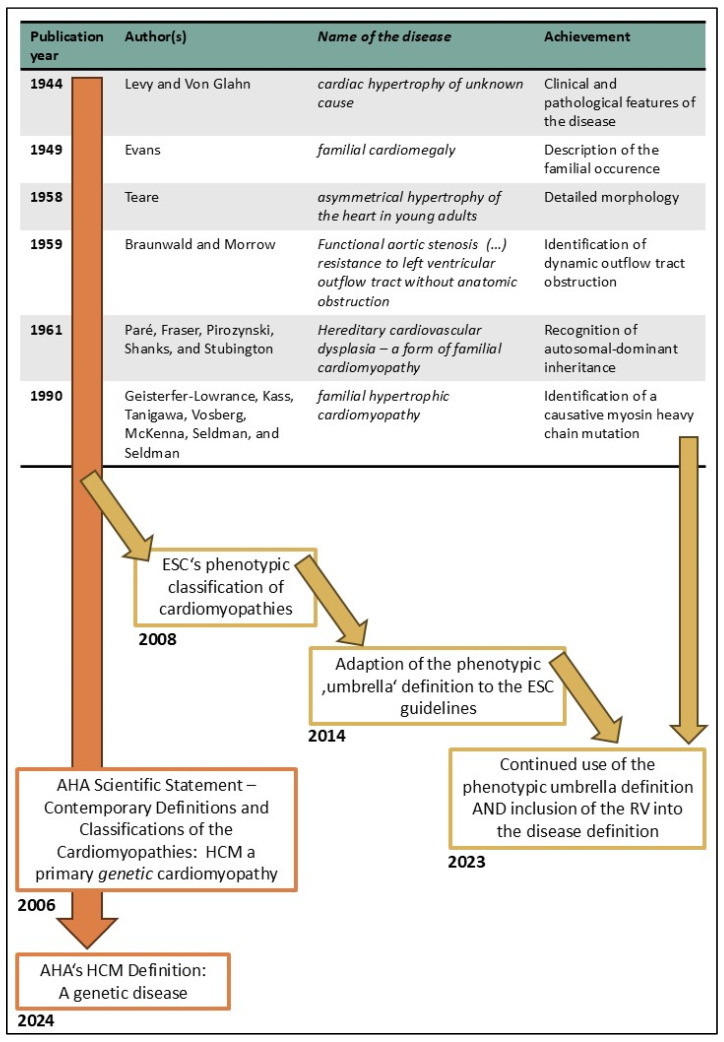
Illustrated is a brief summary of the history of the HCM and how its definitions developed. Additionally, it shows which major classifications build the basis for the current HCM definitions provided by the AHA (orange) and the ESC (yellow). The outlined history follows the summary provided by Braunwald [2]. Other references underlying the figure: [1,3,4,5,6,7,8,10,11,16,18,19,23,24]. Abbreviations: AHA—American Heart Association; ESC—European Society of Cardiology; HCM—hypertrophic cardiomyopathy; RV—right ventricle.

## 4. Pathology of the Thick Heart

Morphology is the starting point for the clinical diagnosis of HCM [7,8]. In adults, a diagnosis is based on a recorded end-diastolic wall thickness of 15 mm or above anywhere within the LV, provided no other causes of hypertrophy are identified [7]. In children, the diagnosis is based on comparing their heart wall thicknesses using z-scores with normal values adjusted for body surface area [7]. HCM is typically diagnosed if the thickness exceeds 2–2.5 standard deviations above the average for children with the same body surface area (z-scores above 2–2.5) [7].

### 4.1. Interventricular Septum

Morphology is also useful in describing the characteristics of HCM [25,26,27], especially regarding the frequently affected IVS [12,25,26,27]. The distribution and shape of myocardial thickening can provide insights into the underlying etiology. For example, a ‘reversed curved’ IVS is more likely to be associated with an HCM-related gene mutation compared to a ‘sigmoid’ septum [26,27]. However, there is no standardized definition for these morphologies, and several terms are used to describe a septum that curves towards the LVOT, such as ‘sigmoid septum’ [28], ‘bulging subaortic septum’ [29], ‘septal bulge’ [30], or ‘angulated septum’ [31].

### 4.2. Apex

Besides the IVS, the apex of the heart is important in HCM morphology, especially in the context of altered myocardial strain patterns. The so-called ‘apical sparing’ may suggest cardiac amyloidosis [32], although it is not pathognomonic for the diagnosis [33]. Additionally, recognizing anomalies in the apex can help detect apical HCM, a subtype more common in Asia [34]. This HCM subform is often associated with both mitochondrial and nuclear deoxyribonucleic acid (DNA) mutations [35]. Moreover, non-compaction cardiomyopathy, another cause of a thickened and spongiform myocardium, often affects the apical portions of the heart [36].

### 4.3. LV Cavity

In addition to changing the appearance of the myocardium and the shape of other cardiac structures (such as papillary muscles and mitral valve [37]), myocardial hypertrophy also alters the geometry of the ventricular cavity [38]. In some instances, these changes in the geometry of the LV cavity can be detected before hypertrophy becomes apparent [38]; the development of a banana-shaped LV is an example [38].

### 4.4. Right Ventricle

It was recognized quite early that the RV is also affected by the HCM. For example, Teare (1958) [4] as well as Braunwald and Morrow (1959) [5] described RV thickening. In 1964, Braunwald et al. [10] and Taylor et al. [23] already described RV obstruction. Nevertheless, a focus on the LV was developed, for example, in the HCM definitions applied by the AHA currently [7,15] or ESC in the past [16]. Interestingly, the attention is now shifting to the RV again: the recent 2023 ESC guidelines incorporate the RV into the HCM definition [8]. Systematic studies have revealed that RV involvement in HCM is variable. RV hypertrophy is estimated to occur in 28–44% of patients and often co-exists with LV hypertrophy [39]. On the one hand, the RV can be the primary site [40,41]. On the other hand, RV hypertrophy can be secondary, such as in pulmonary hypertension or LV disease with increased RV afterload manifesting as enhanced diastolic stiffness, collagen accumulation, or sarcomere stiffening in the RV wall [39]. The spectrum of RV involvement ranges from mild thickening due to LV hypertrophy [42] to more clinically relevant RVOT obstruction [41]. While overt RV systolic dysfunction is generally discrete, alterations in RV longitudinal strain are pronounced, suggesting early impairment [43]. There are cases of sole or primary RV involvement [40,41]. When affected, the RV shares histological similarities with the LV, including cardiomyocyte disarray and fibrosis [44]. However, the genetic underpinnings driving RV changes remain inadequately described. Morphological assessments have been somewhat limited, but earlier studies suggest that the RV in HCM may adopt a more globular configuration with a shorter apex-to-base ratio relative to that in healthy individuals [45].

### 4.5. Time and Site of Onset of Hypertrophy

HCM-associated mutations are congenital, but hypertrophy usually manifests during young adulthood [46], and childhood-onset is rather uncommon [47]. Nevertheless, neonatal ventricular hypertrophy can occur in cases of neonatal hyperinsulinism in general [48] but is often related to maternal diabetes [49,50]. This can present as LV, RV, or biventricular hypertrophy [51]. In older adults, the onset of hypertrophy can be attributed to several diseases leading to biventricular hypertrophy [52]. For instance, arterial hypertension can lead to concentric LV hypertrophy [53] or isolated RV hypertrophy in individuals aged around 60 years [54]. Older adults and obese individuals with hypertension are prone to biventricular hypertrophy [55].

Beyond age-related conditions, hypertrophy can also be linked to exercise at any age, with physical activity resulting in different morphologies of the ‘Athlete’s Heart’ [56]. Endurance training seems to be associated with larger LV cavities, whereas strength training tends to result in thicker ventricular walls [56]. Intensive training can even lead to HCM-like hypertrophy [57]. Exercise-induced hypertrophy is generally less pronounced in pediatric populations than in adults [58]. Differentiating between the athlete’s heart, a physiological hypertrophic remodeling, and HCM as a pathological maladaptive hypertrophic response remains a diagnostic challenge [56,59]. Considering the cavity size, as well as a detailed clinical examination and anamnesis (i.e., exploring the familial history), can be helpful in distinguishing both entities [59].

From a morphological point of view, hypertrophy can be described based on the affected ventricle, such as LV, RV, or biventricular (for example, reference [51]). In conjunction with clinical features like age and accompanying diseases, the distribution of hypertrophy can provide diagnostic clues [60]. For example, the AHA states that HCM usually affects the LV in younger adults [16]. In contrast, biventricular hypertrophy in older adults can be suggestive of cardiac amyloidosis [61].

### 4.6. Ventricular Obstruction

When hypertrophy causes obstruction of the ventricular cavity or OT, the condition is termed hypertrophic obstructive cardiomyopathy [37] or, more recently, obstructive HCM [62,63]. This usually dynamic stenosis of the LVOT during the systole is an important pathophysiological feature [64]. It increases myocardial workload and energy demand through the Anrep effect [65] and can even lead to mitral regurgitation due to the systolic anterior motion of the mitral valve apparatus (the so-called ‘SAM phenomenon’) [64]. This further increases both the volume and pressure load on the already impaired ventricle [66]. Obstruction can occur in both chambers of the heart and at various levels within each ventricle. For example, LV obstructions can occur at a subaortic [26], mid-ventricular [67,68], or mid-apical level [69]. Several conditions besides AHA-defined HCM [7,15], such as chronic systemic hypertension, can also cause LV obstruction [68]. Concerning the RV, congenital (or acquired) malformations such as the double-chambered RV can be associated with progressive RV mid-ventricular obstruction [70]. Further, HCM or aortic root aneurysms can cause RVOT obstructions in adults [71]. RV or LV hypertrophy can also occur in pulmonary or aortic valve stenoses, respectively. Knowing the obstruction site is of particular importance when planning therapeutic approaches to relieve the obstruction [69].

### 4.7. Causes of Ventricular Hypertrophy

In the context of HCM definitions, attention should be given to the cause of ventricular wall thickening. In AHA-defined HCM, both myocardial and endocardial thicknesses typically increase [72,73], with myocardial thickening being more predominant. This thickening [7,16,72,73] mainly arises from cardiomyocyte hypertrophy, leading to a thick and solid ventricular wall [68]. Other entities associated with a thickened heart show different properties of the ventricular wall. For example, non-compaction cardiomyopathy manifests in an increase in the overall wall thickness and a typical two-layered ventricular wall with a spongiform inner myocardium [74]. Similarly, cardiomyocytes may present with spongiform features, such as intracellular vacuoles observed in glycogen storage diseases [75] or Fabry disease [76]. These vacuoles, seen as empty spaces in standard stains, are artifacts of histological tissue processing and have shown varying content (such as mitochondria or glycogen) with different underlying diseases in electron microscopy studies [77,78,79]. The ESC definition of HCM encompasses not only diseases with varying phenotypes but also genotypes; for instance, genetic diseases associated with myocardial hypertrophy are included, although they do not involve mutations of genes encoding sarcomeric or sarcomere-associated genes [8,16]. A notable example is Barth syndrome, an X-linked inherited disorder caused by mutations in the TAFAZZIN gene encoding a mitochondrial transacylase involved in the biogenesis of cardiolipin (a mitochondrial-specific phospholipid) [80]. Barth syndrome can manifest with myocardial hypertrophy and dilative cardiomyopathy and is sometimes accompanied by non-compaction [80], with reports documenting varying cardiac phenotypes within a single family [80]. With the ESC ‘umbrella’ [19] definition, not all cases of ventricular wall thickening involve true hypertrophic remodeling and may, therefore, not accurately represent HCM in the authors’ opinion. Examples include transient forms of myocardial thickening like myocarditis [81] and extra-myocardial diseases like endocardial fibroelastosis that can also cause ventricular wall thickening [82]. Figure 2 summarizes some relevant aspects of differential diagnosis for pathological thickening of the ventricular wall.

Regarding the cause of ventricular hypertrophy, the classification of cardiomyopathies is generally heterogeneous. The AHA advocates a differentiation between primary and secondary cardiomyopathies [24]. Primary cardiomyopathies primarily affect the heart muscle and may have genetic, acquired, or mixed causes [24]. In contrast, secondary cardiomyopathies include myocardial involvement as part of a broader spectrum of pathological findings [24]. However, such a differentiation can be challenging, as primary cardiomyopathies can also have extra-cardiac manifestations [18]. Therefore, the phenotypic approach has been adopted in Europe [18]. In 2008, this approach differentiated between familial and non-familial and genetic and non-genetic forms of HCM, dilated cardiomyopathy, arrhythmogenic right ventricular cardiomyopathy, restrictive cardiomyopathy, and unclassified cardiomyopathies [18]. The most recent ESC guidelines from 2023 distinguish between HCM, dilated cardiomyopathy, non-dilated left ventricular cardiomyopathy, arrhythmogenic right ventricular cardiomyopathy, and restrictive cardiomyopathy based on morphological and functional clinical characterization [8].

## 5. The Definitional Gap for HCM

The different definitions of HCM lead to significant variations in what constitutes HCM and the pathological aspects that should be considered. Figure 3 summarizes the key differences between the two definitions, while Table 1 provides examples of how the different definitions affect different pathological key factors such as etiology, prevalence, time of onset, and so forth.

**Figure 2 diagnostics-14-02534-f002:**
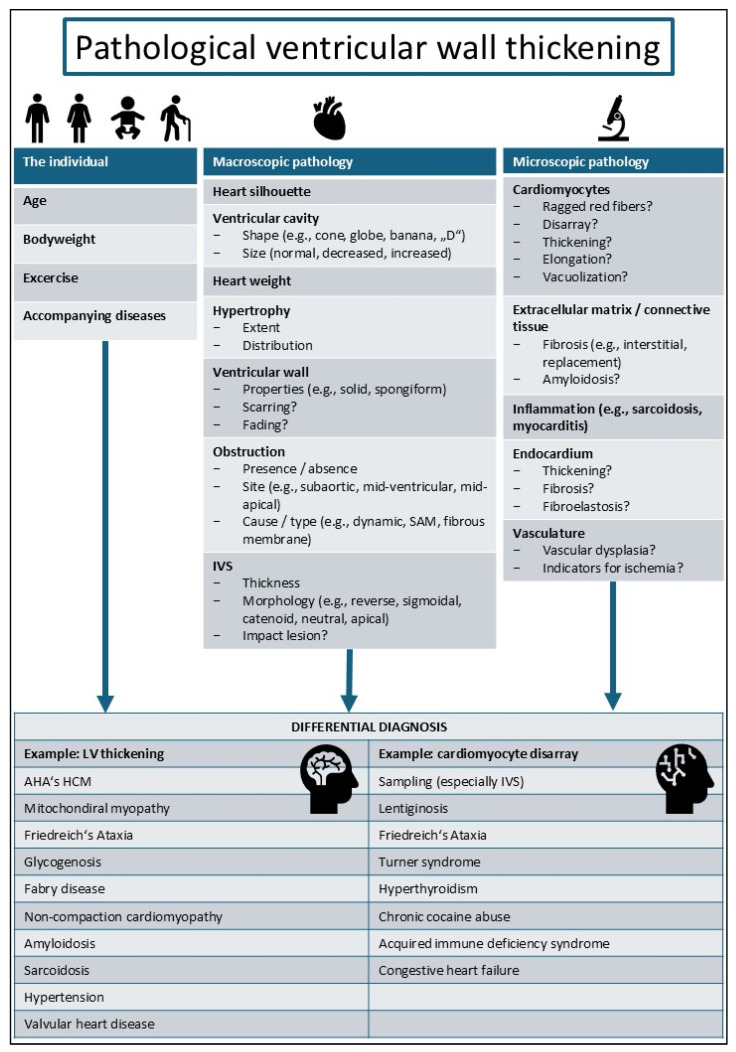
A brief summary of pathoanatomical aspects of thickened myocardium and its differential diagnostic implications. The basic idea of the figure was inspired by the textbook Pathology of the Heart and Sudden Death in Forensic Medicine [83]. Other references underlying the present figure [26,27,37,38,60,72,73,74,84,85,86,87,88]. Abbreviations: AHA—American Heart Association; HCM—hypertrophic cardiomyopathy; IVS—interventricular septum.

**Figure 3 diagnostics-14-02534-f003:**
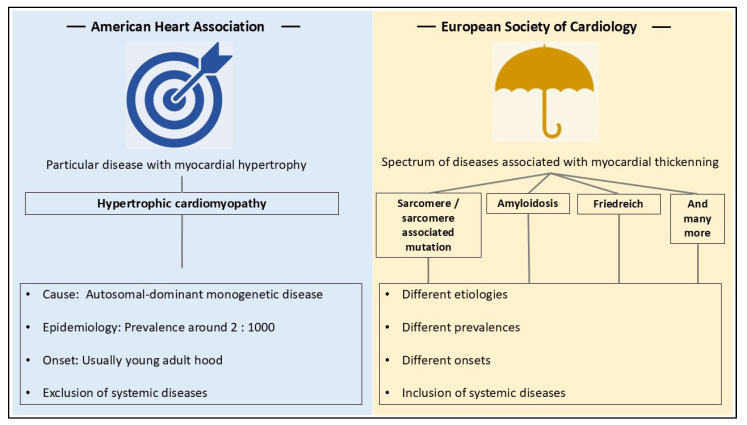
Differences between the available major definitions of HCM provided by the ESC [8] and the AHA [7]. The term ‘umbrella diagnosis’ for the ESC definition of HCM was already introduced by Lillo et al. [19]. Abbreviations: AHA—American Heart Association; ESC—European Society of Cardiology; HCM—hypertrophic cardiomyopathy.

HCM was first described in the 1940s [1,2,3], and it was eventually identified as an inherited [6] genetic [11] disease. Thus, the AHA definition aligns more closely with the foundational knowledge and the trajectory of basic research understanding (Figure 1). In contrast, the ESC phenotypic approach, introduced in 2008 [18] and included in the guidelines in 2014 [16], appears to depart from this accumulated knowledge (Figure 1). Historically, most of the scientific understanding of HCM has been generated using models of sarcomeric mutations, which are consistent with the AHA definition. A key example is the extensive research on the arrhythmogenicity of HCM [89,90,91]. As outlined below and in Table 1, the definitional gap presents additional hurdles, such as complicating the interpretation of epidemiological data, especially regarding the age of onset and the risk of SCD in different populations.

### 5.1. Prevalence

The AHA defines the HCM as a genetic autosomal-dominant disorder [7] and identifies it as the most frequent hereditary cardiac disease [92]. The reported prevalence varies, ranging from 2 of 1000 young adults [93] to 1:344 or 1:625 cases [12]. Approximately 30–60% of persons with AHA-defined HCM have pathogenic mutations [7]. In contrast, the ESC definition indicates that 60% of adolescents and adults diagnosed with HCM have autosomal dominant mutations in cardiac sarcomere protein genes [16]. The broader, umbrella-like ESC definition [19] implies a higher prevalence of HCM, which raises questions about the comparability of epidemiological data, such as age-related data. For instance, cardiac amyloidosis is typically a disease of older adults, whereas AHA-defined HCM is often associated with SCD in the young [94] or in athletes [59]. This issue becomes significant when epidemiological studies are published in journals not affiliated with one of the major cardiology societies, especially if they do not clearly specify which HCM definition the study follows (see examples [95,96]).

**Table 1 diagnostics-14-02534-t001:** Differences resulting from the different HCM definitions.

Clinical/Pathological Feature	AHA-Defined HCM	ESC-Defined HCM
Inheritance	Autosomal dominant [7].	Barth syndrome: X-linked [80]. AHA-defined HCM: Autosomal dominant [7]. Noonan syndrome: Autosomal dominant [97]. Fabry disease: X-linked [98]. Glycogen storage disease type IIIa: Autosomal recessive [99]. etc.
Etiology	Monogenetic disorder caused by a sarcomeric or sarcomere-associated mutation [7,92].	Cardiac Amyloidosis: Light chain cardiac amyloidosis—bone marrow disease [100]; Transthyretin amyloidosis—hereditary or senile [100]. Cardiac Sarcoidosis: Response of the immune system to an unknown antigenic trigger [101]. Barth syndrome: Mutation of the TAFAZZIN gene [80]. AHA-defined HCM: Monogenetic disorder caused by a sarcomeric or sarcomere-associated mutation [7,92]. etc.
Shape IVS	Can give hints regarding the genetic background [26,27].	AHA-defined HCM: Can give hints regarding the genetic background [26,27]. Noonan syndrome: The authors could not identify a study on the diagnostic implications of IVS shape in this disease. Cardiac amyloidosis: Can also exhibit sigmoid and reverse shape [102]. A study showing particular diagnostic implications of the IVS shape in amyloidosis was not observed by the authors. etc.
Onset	Usually in young adulthood [46]. Rarely in childhood [47].	Neonatal ventricular hypertrophy: Maternal diabetes [49,50]; Hyperinsulinism in general [48]. Cardiac amyloidosis: Often patients older than 65 years [100]. Cardiac sarcoidosis: Mostly in individuals from 25 to 60 years [101]. etc.
Mortality	An annual mortality rate of up to 6% has been reported [103]	The ESC guidelines do not report a specific HCM mortality rate [8,16]. Cardiac amyloidosis: Majority of affected individuals die of cardiovascular causes; 5-year cumulative proportion of mortality ranges from 44% to 65% [104]. Noonan syndrome: Mortality rate of 9% with an age of death ranging from some months to around 60 years [97]. etc.
Prevalence	In the range of 1:500 [7].	AHA-defined HCM: In the range of 1:500 [7]. Noonan syndrome: Estimated prevalence 1 in 1000–2500 [97]. Cardiac amyloidosis: Varying reports, for example, less than 1 in 2000 European patients versus 16% of the patients with degenerative aortic stenosis and 13–17% of those with heart failure with preserved ejection fraction [100]. etc.

The table provides examples of gaps resulting from the differing HCM definitions for clinical and pathological key features. Further details are provided in the main text of the manuscript. Abbreviations: AHA—American Heart Association; ESC—European Society of Cardiology; HCM—hypertrophic cardiomyopathy; IVS—interventricular septum.

### 5.2. Mortality

In individuals diagnosed based on the AHA HCM criteria, SCD emerges as the primary concern, especially among young and asymptomatic individuals [105]. Extreme forms of hypertrophy consistently carry a higher risk of SCD [106,107]. Additionally, sex influences the risk of cardiac death, as women with ventricular hypertrophy face a higher relative risk for cardiac death than men [108]. This sex disparity is also evident in animal models, where male and female mice with HCM show varied exercise adaptations [109]. Other factors, such as OT obstruction, have not been consistently linked to the increased risk of SCD [107,110], although ventricular obstruction at rest has been identified as a potential risk factor, especially in athletes [111].

For AHA-defined HCM, an annual mortality rate of up to 6% has been reported [103]. In contrast, the broader ‘umbrella’ [19] definition of the ESC does not provide a specific mortality rate for this disease spectrum [8,16]. Some recent studies on HCM mortality acknowledge its hereditary nature but do not clarify whether they base their analysis on the AHA or ESC HCM definition [112]. This raises a critical question: Are these mortality data specific to a particular sarcomeric or sarcomere-associated disease, or do they reflect cumulative mortality across a spectrum of diseases?

Taken together, the AHA HCM definition allows for a more precise determination of mortality rates. For the various diseases grouped under the broader ESC HCM definition, specific mortality rates can be reported, such as those for cardiac amyloidosis or cardiac sarcoidosis. From the perspective of a forensic pathologist, determining the exact cause of death (such as AHA-defined HCM, myocarditis, or another condition) is essential for accurate mortality rate estimation and risk stratification. This precision may explain why consensus statements in the post-mortem sector on SCD cases gravitate toward the AHA definition [113].

### 5.3. Basic Science Implications

The different HCM definitions also create challenges in basic science research. For example, Barth syndrome is referred to as HCM under the ESC ‘umbrella’ [19] definition by some research groups [114], whereas other European groups describe it as both an HCM phenotype and an independent entity [115]. The European consortium adheres to the AHA definition of HCM [115]. Without careful attention to the specifics of each study, researchers may inadvertently apply knowledge from unrelated diseases, making hypothesis generation and experimental design challenging.

### 5.4. Myosin Inhibition—A Comprehensive Therapeutic Approach

The profound influence of basic science is exemplified by the development of myosin inhibitors. AHA-defined HCM is characterized by myocardial hypercontractility [116,117], which is driven by increased myocardial activation at low (diastolic) calcium concentrations and higher activity of the myosin-ATPase [118,119]. This hypercontractile state is in part dependent on the mobilization of ‘dormant’ myosin molecules previously in an energy-conserving (super-relaxed) state. Given the central role of myosin in the pathophysiology of HCM, it has become a focal point in treatment research, as demonstrated by the myosin inhibitor mavacamten [120]. Mavacamten reversed the hypercontractility, maladaptive cardiac remodeling, myofibrillar disarray, and fibrosis in HCM animal models [121]. In patients with obstructive HCM, mavacamten reduced LVOT obstruction, improved diastolic function, reduced symptoms, augmented exercise capacity, and was associated with decreased expressions of biomarkers of cardiac stress and injury [120,122,123,124]. If mavacamten proves effective for ESC-defined hypertrophic myocardium, it may suggest a unified mechanism underlying myocardial hypertrophy, irrespective of its initial upstream triggers. Additionally, emerging insights into the role of epigenetics in hypertrophy and heart failure [125] may help to better characterize the different subsets of hypertrophic myocardium, regardless of how they are termed, classified, or defined. Such increasing understanding of the disease may also clarify why only 30–60% of individuals with AHA-defined HCM have the defining pathogenic or likely pathogenic mutations [7] and why they exhibit myocardial hypertrophy.

## 6. Summary and Conclusions

The definitions of HCM put forward by the ESC and the AHA reflect two distinct perspectives. The ESC adopts an ‘umbrella’ approach [19] that characterizes HCM as a ‘spectrum of the thick heart’. The AHA defines it as a specific genetic disorder. Both definitions offer unique advantages. The broader phenotypic approach by the ESC aims to facilitate diagnosis, which makes it more practical and inclusive [16]. This definition includes a wider range of individuals, including those who meet the AHA criteria for HCM but lack the typical sarcomere (and sarcomere-associated) mutations. However, this broad categorization is associated with the risk of diluting decades of accumulated knowledge on the morphological and functional characteristics of HCM, especially when interpreting epidemiological and mortality data. Therefore, it is essential to critically assess HCM-related research within the context of the definition applied. In the experience of the authors, explicitly stating the HCM definition applied is crucial for accurate planning and execution of research projects. Given that the ESC has introduced several new terms for the classification of cardiomyopathies [8,18], the suitability of a new term, such as ‘myocardial hypertrophy-associated cardiomyopathy’, to distinguish this expansive approach from the more traditional definition remains to be explored.

While conducting this review, the authors observed that many studies did not specify which HCM definition was applied. On one hand, this emphasizes that there is limited awareness regarding the difference between the HCM definitions. On the other hand, this did not allow us to analyze in detail which differences between different studies are attributable to the difference between the definitions and how the definition applied influenced the study design.

A unified understanding of the HCM spectrum and its various forms of hypertrophic remodeling, however, is fundamental to the collaborative efforts of clinicians, post-mortem specialists, and basic scientists. Achieving consensus on key features, such as terms describing IVS morphology and ventricular shape, will significantly enhance interdisciplinary research efforts. By addressing the challenges arising from these differing definitions, the authors hope to contribute meaningfully to the ongoing scientific dialogue on HCM. In their view, this ongoing discussion is essential for developing a more nuanced, effective, and comprehensive understanding of HCM.

## Data Availability

The present manuscript resembles a narrative review. The literature search performed is detailed below in Appendix A.

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
