# Peer review of "Definitional Challenges in Understanding Hypertrophic Cardiomyopathy"

_diagnostics, 2024, doi:10.3390/diagnostics14222534_

Round 1

Reviewer 1 Report

Comments and Suggestions for Authors

To conduct the review, a review of the literature is presented; it is devoted to the current problem of cardiovascular diseases - hypertrophic cardiomyopathy

The review represents a non-standard approach; it compares two concepts of hypertrophic cardiomyopathy: the American one, based on the idea of ​​this pathology as a hereditary genetic one, and the European one, based on identifying the phenotype of myocardial hypertrophy with different etiologies A large number of modern studies and publications have been analyzed, as well as historical background has been presented. Of interest are tables and figures that summarize the etiology of myocardial hypertrophy The authors express their opinion on this issue, based on the real contribution of these two ideas both to stratify the prognosis and statistics for this pathology The authors conclude that it is possible to use a new term, such as ‘myocardial hypertrophy-associated cardiomyopathy,’ to distinguish this expansive approach from the more traditional definition remains to be explored.

The review is of scientific and practical interest and can be published

Author Response

Comment 1: To conduct the review, a review of the literature is presented; it is devoted to the current problem of cardiovascular diseases - hypertrophic cardiomyopathy. The review represents a non-standard approach; it compares two concepts of hypertrophic cardiomyopathy: the American one, based on the idea of this pathology as a hereditary genetic one, and the European one, based on identifying the phenotype of myocardial hypertrophy with different etiologies A large number of modern studies and publications have been analyzed, as well as historical background has been presented. Of interest are tables and figures that summarize the etiology of myocardial hypertrophy The authors express their opinion on this issue, based on the real contribution of these two ideas both to stratify the prognosis and statistics for this pathology The authors conclude that it is possible to use a new term, such as ‘myocardial hypertrophy-associated cardiomyopathy,’ to distinguish this expansive approach from the more traditional definition remains to be explored. The review is of scientific and practical interest and can be published.

Reply 1: We thank the reviewer for the encouraging feedback and are delighted that our manuscript was well-received. In addition to the changes suggested by Reviewer 2 (see below), we made further adjustments to improve the clarity of abbreviations and refined the figures and table for better clarity.

Reviewer 2 Report

Comments and Suggestions for Authors

This manuscript addresses the challenges associated with differing definitions of hypertrophic cardiomyopathy (HCM) from a pathoanatomical perspective, with a particular focus on the discrepancies between the definitions provided by the European Society of Cardiology (ESC) and the American Heart Association (AHA). The manuscript provides a detailed discussion of how these definitional differences impact research outcomes and clinical applications, contributing significantly to the understanding of HCM.

The manuscript systematically discusses the variations in HCM definitions with a clear research focus. The discussion from a pathoanatomical viewpoint is thorough and helps the reader understand the complex characteristics of HCM. Furthermore, the historical background and comparison of current clinical guidelines from the ESC and AHA are well presented.

Definitional discrepancies in HCM are highly relevant and critical for understanding the condition’s implications for clinical studies and statistical data interpretation. This manuscript offers a unique perspective on how this gap affects HCM diagnosis and treatment, fostering a shared understanding among professionals.

The manuscript effectively utilizes various references to explore the issues surrounding HCM definitions. The pathoanatomical approach is well supported by a comprehensive literature review, making it valuable for basic researchers and clinicians alike.

Suggestions for Improvement

Table 1 and Fig.2 option 2 need to be understood easily.

Please rewrite the table and make the figure more straightforward.

Providing more case examples illustrating how the ESC and AHA definitions differ in specific scenarios could add depth, especially regarding how each definition impacts patient diagnosis and prognosis.

Including additional tables or figures summarizing key differences and their impacts would help enhance reader comprehension.

This manuscript highlights critical issues regarding the definitional gap in HCM and provides valuable insights from a pathoanatomical perspective. Considering the suggested improvements, the manuscript offers substantial value and is recommended for publication.

Author Response

Reviewer comment 1: This manuscript addresses the challenges associated with differing definitions of hypertrophic cardiomyopathy (HCM) from a pathoanatomical perspective, with a particular focus on the discrepancies between the definitions provided by the European Society of Cardiology (ESC) and the American Heart Association (AHA). The manuscript provides a detailed discussion of how these definitional differences impact research outcomes and clinical applications, contributing significantly to the understanding of HCM. The manuscript systematically discusses the variations in HCM definitions with a clear research focus. The discussion from a pathoanatomical viewpoint is thorough and helps the reader understand the complex characteristics of HCM. Furthermore, the historical background and comparison of current clinical guidelines from the ESC and AHA are well presented. Definitional discrepancies in HCM are highly relevant and critical for understanding the condition’s implications for clinical studies and statistical data interpretation. This manuscript offers a unique perspective on how this gap affects HCM diagnosis and treatment, fostering a shared understanding among professionals. The manuscript effectively utilizes various references to explore the issues surrounding HCM definitions. The pathoanatomical approach is well supported by a comprehensive literature review, making it valuable for basic researchers and clinicians alike.

Suggestions for Improvement: Table 1 and Fig.2 option 2 need to be understood easily. Please rewrite the table and make the figure more straightforward. Providing more case examples illustrating how the ESC and AHA definitions differ in specific scenarios could add depth, especially regarding how each definition impacts patient diagnosis and prognosis. Including additional tables or figures summarizing key differences and their impacts would help enhance reader comprehension.

Response 1: We thank the reviewer for the thoughtful suggestions. We have made the following revisions to our figures and tables for greater clarity: positive feedback. Below you will find an overview of how we have implemented your suggestions for improvement.

  • Removed former Figure 2: To streamline the manuscript and avoid unnecessary redundancy, we removed the previous Figure 2 (which illustrated the overlap between ESC and AHA definitions), as this discussion is already adequately covered in the text. We revised the wording in paragraph 5.4 accordingly (page 11 in simple markup).
  • Added historical timeline Figure: We added a new figure to depict the historical course of HCM research and definitions, outlining key pathologic criteria established by both major societies. This is now Figure 1 (page 3 in simple markup).
  • Transformed Table 1 into a Figure: Recognizing that the original Table 1 was unclear, we transformed it into a figure (now Figure 2, page 7 in simple markup) for improved visual clarity and ease of understanding.”
  • Provided comparative examples: We agree that more examples of how the differing definitions interfere with important measures, such as prevalence and mortality, would be beneficial. Therefore, we added a comparative table (new Table 1, can be found on page 9 in the simple markup).
  • Added references to Figures and Table: To improve transparency, we included references in all figure legends and the new table.

Reviewer comment 2: This manuscript highlights critical issues regarding the definitional gap in HCM and provides valuable insights from a pathoanatomical perspective. Considering the suggested improvements, the manuscript offers substantial value and is recommended for publication.

Response 2: We thank the reviewer again for the positive feedback and hope that our revisions adequately address your concerns.